# Cereal Waste Valorization through Conventional and Current Extraction Techniques—An Up-to-Date Overview

**DOI:** 10.3390/foods11162454

**Published:** 2022-08-14

**Authors:** Anca Corina Fărcaș, Sonia Ancuța Socaci, Silvia Amalia Nemeș, Liana Claudia Salanță, Maria Simona Chiș, Carmen Rodica Pop, Andrei Borșa, Zorița Diaconeasa, Dan Cristian Vodnar

**Affiliations:** 1Department of Food Science, Faculty of Food Science and Technology, University of Agricultural Sciences and Veterinary Medicine of Cluj-Napoca, 3–5 Mănăştur Street, 400372 Cluj-Napoca, Romania; 2Department of Food Engineering, Faculty of Food Science and Technology, University of Agricultural Sciences and Veterinary Medicine of Cluj-Napoca, 3–5 Mănăştur Street, 400372 Cluj-Napoca, Romania; 3Laboratory for Testing Quality and Food Safety, Calea Florești Street, No. 64, 400516 Cluj-Napoca, Romania; 4Institute of Life Sciences, University of Agricultural Sciences and Veterinary Medicine, Calea Mănăștur, 400372 Cluj-Napoca, Romania

**Keywords:** cereal by-products, advanced extraction techniques, bioactive fractions, circular bioeconomy

## Abstract

Nowadays, in the European Union more than 100 million tons of food are wasted, meanwhile, millions of people are starving. Food waste represents a serious and ever-growing issue which has gained researchers’ attention due to its economic, environmental, social, and ethical implications. The Sustainable Development Goal has as its main objective the reduction of food waste through several approaches such as the re-use of agro-industrial by-products and their exploitation through complete valorization of their bioactive compounds. The extraction of the bioactive compounds through conventional methods has been used for a long time, whilst the increasing demand and evolution for using more sustainable extraction techniques has led to the development of new, ecologically friendly, and high-efficiency technologies. Enzymatic and ultrasound-assisted extractions, microwave-assisted extraction, membrane fractionation, and pressure-based extraction techniques (supercritical fluid extraction, subcritical water extraction, and steam explosion) are the main debated green technologies in the present paper. This review aims to provide a critical and comprehensive overview of the well-known conventional extraction methods and the advanced novel treatments and extraction techniques applied to release the bioactive compounds from cereal waste and by-products.

## 1. Introduction

Cereals are one of the most significant crops and food sources in the human diet. The cereals processing chain generates huge amounts of agricultural waste, which is known as lignocellulosic biomass. It is estimated that 12.9% of all food wastes are produced during cereal processing and manufacturing, and 30% of the cereal weight basis is lost or wasted [1,2].

Depending on the type of cereal and on the process applied, cereal waste by-products are generated in different technological steps: straw and cob are the wastes produced during the cleaning grain process, bran is generated when the cereal is used for the extraction of some compounds, such as corn and rice oil, or when the grains are polished during the rice processing [2]. Parboiling rice generates a large amount of waste which can be used as a biofuel, and the beer industry produces brewery spent grain (BSG), a waste containing phenolic compounds with antioxidant and antimicrobial properties [2,3]. Wheat bran, wheat germ, rice bran, rice germ, corn germ, corn bran, barley bran, and brewery spent grain (BSG) are just a few examples of wastes that may be exploited to recover bioactive compounds and therefore promote a sustainable approach for the development of novel food products and ingredients. “Zero waste economy” is a new eco-innovative concept which is based on the use of waste like a raw material for new products and applications [4]. 

Brewers’ spent grain (BSG) is a by-product of the industrial brewing process mostly used in animal feed. The insoluble fraction of barley grains separated before the wort fermentation process represents approximately 85% of the total by-products of the brewing process [5]. In 2017, European beer production was approximately 39.7 billion liters, with BSG accounting for 20 kg every 100 L of beer [6], whilst, in 2019, a total amount of 1.91 billion hectoliters of beer were produced worldwide [7]. BSG is mainly composed of a lignocellulosic substrate that is rich in proteins, polysaccharides, and bioactive compounds including polyphenols [6]. On the other side, rice bran is a rich source of minerals, crude fibers, proteins, or even vitamins. It is worthy of note to also mention its lipids content with high antioxidant compounds such as tocotrienols, oryzanols, and tocophenols [2].

Food industry wastes come out as potential valuable protein sources to be valorized; thus, cereal wastes are mainly used due to their immense release rates and high protein contents compared to other by-products. High-value compounds such as lignans, essential fatty acids, ferulic acid and phenols, tocopherols, anthocyanins, or ß-glucans are found in wheat, rice, corn, or barley wastes. After extraction, they are used in the food industry for the fortification of different types of food (bread, pasta, biscuits), as natural colorants, or in the pharmaceutical and cosmetic industries as a moisturizing agent [8]. The recovery of bioactive fractions by conventional and modern technologies and their further valorization in different industries is illustrated in Figure 1.

The most crucial stage in isolating various types of bioactive molecules from cereals is the extraction process. Bioactive compounds have been extracted using both conventional and innovative extraction protocols from cereal waste such as pressurized liquid extraction of brewer’s spent grain [9], microwave-assisted extraction of rice bran [10], supercritical fluid extraction, pressurized liquid extraction, enzyme-assisted extraction from buckwheat hulls [11], enzyme- and ultrasound-assisted extraction of sesame bran [12], enzymatic hydrolysis of soy bean [13], combined hydrothermal pre-treatment and enzymatic hydrolysis of corn fiber [14], and enzymatic extractions of barley bran [15].

Conventional extraction technologies such as organic solvent-based extraction, maceration, and hydrodistillation are mainly used for a long period of time in the food industry to extract nutritionally essential and non-essential components. According to a large body of literature, the use of conventional methods has some disadvantages, as follows: high cost, high purity solvents, low extraction yield, time-consuming, large amount of reagents, and low purity extracts [16,17,18]. Considering the mentioned weaknesses of the conventional extraction methods, intensive efforts have been made for the development of more efficient, sustainable, and ecologically friendly extraction technologies, such as enzyme-assisted extraction, supercritical fluid extraction, microwave-assisted extraction, and ultrasound-assisted extraction [17].

The aim of this review is to analyze comprehensively both conventional extraction techniques such as acid and alkaline hydrolysis, solvent extraction, and Soxhlet extraction, along with the advanced novel treatments and extraction techniques (enzyme-assisted extraction, ultrasound-assisted extraction, microwave-assisted extraction, membrane technology, subcritical and supercritical extraction, pressurized liquid extraction, steam explosion, pulsed electric field, and high voltage electrical discharge).

## 2. Conventional Extraction Methods

Several conventional extraction methodologies have been reported for the recovery of phenolic compounds from agro-food waste. The main variables that influence the working extraction method are based on the cost, time, and availability of the process [19]. However, there is an increasing need for green and sustainable approaches leading to phenolic-rich extracts with low environmental impact [20].

### 2.1. Acid and Alkaline Hydrolysis

The alkaline hydrolysis method is frequently used to cleave the lignin/phenolic-carbohydrate complexes structure, resulting in phenolic compounds, soluble sugars, insoluble lignin, and carbohydrates [21]. This method can be performed in auto-pressurized tubes or cylindrical stainless-steel reactors at high temperatures and high pressures [21]. Alkaline reagents act by disturbing the cell wall dissolving lignin and hemicelluloses and releasing ferulic and *p*-coumaric acids [22]. On the other hand, Burlini et al. [23] showed that alkaline hydrolysis could also be successfully used for the extraction of ferulic acid from yellow and white maize germ.

However, the extraction procedure can present several disadvantages such as an incomplete extraction of the bounded antioxidants, a low extraction yields due to the antioxidant solubility, and the loss of synergetic effect between antioxidants. Moreover, the phenolic compounds could be further degraded and oxidized during the alkali extraction [24]. Table 1 presents an overview of the main phenolic compounds extracted by using conventional extraction techniques.

### 2.2. Solvent Extraction

The extraction of soluble phenolic compounds is the initial step throughout every application. The structure and polarity of these compounds affect their solubility and therefore the extraction process [37]. Considering varied extraction processes, total antioxidant activities of cereals and grain products may have been underestimated due to the difficulties of solubilizing the phenolic compounds before assessment [24]. The yield of phenolic compound extraction could be positively influenced through the breakdown of cellular structures by pretreatment processes such as maceration, grinding, milling, and homogenization. These actions are designed to enhance the contact area between the solvent and the sample, and therefore, improve the extraction yield [38]. Additionally, to enhance the extraction yield, several solvents such as water, ethanol, methanol, and acetone might be utilized alone or in combination [24].

Organic solvents were traditionally used to extract various phenolic compounds from natural sources. Solid-liquid extraction (SLE) is the extraction technique performed most often [19,38], especially because of its accessibility, efficiency, and wide applicability [38]. The method’s effectiveness is influenced by the required temperature and time, as well as by the solvent composition [19]. This affirmation is sustained by Meneses et al. [35] who evaluated the efficiency of the extraction with different types of solvents (methanol, ethanol, acetone, hexane, ethyl acetate, water, methanol/water mixtures, ethanol/water mixtures, and acetone/water mixtures) aiming to extract the antioxidant phenolic compounds from BSG. Hot water bath and magnetic agitation were used for the extraction of phenolic compounds from BSG samples. The results emphasized that acetone–water mixture at 60% (*v*/*v*) has the highest yield of phenolics (9.90 ± 0.41 mg gallic acid equivalents)/g dw), even if all the extracts showed antioxidant activity and a strong correlation between phenolic, flavonoids, and antioxidant compounds [35].

It can be concluded that the solvent plays a significant role in the extraction of BSG phenolic compounds, highlighting the possibility of using “green solvents” (aqueous acetone or aqueous ethanol) in the extraction of food waste bioactive compounds. This idea is in line with Socaci et al. [33], who demonstrated that the chosen solvent for the BSG phenolic compounds extraction plays a key role in the extraction process, highlighting that the most efficient solvents were ethanol–water and acetone–water mixtures, respectively. In line with this, Guido et al. showed that some phenolic compounds such as cinnamic and benzoic acids, due to their high polarity, could not be completely extracted by using pure solvents, and therefore, the use of mixtures such as acetone–water and alcohol–water are highly recommended [38]. Besides the phenolic compound recovery, the SLE technique has certain drawbacks, such as the high volume of the required solvents and long extraction time [8,38]. The method’s downsides include the restricted use of solvents due to food safety concerns, as well as the risk of target component degrading [38].

Similarly, Smuda et al. attempted to link the quantity of bioactive compounds with the different solvents employed for extraction in the case of total phenols content (TPC) and antioxidant activity of wheat, rice, and maize milling by-products. The use of ethanol resulted in higher TPC and antioxidant activity than methanol, water, or acetone, according to Smuda et al. [34].

Traditional Solvent Extractions (TSEs) were frequently applied to analyze total antioxidant activities in food items. However, their accuracy is controversial due to the numerous differences, such as the variability of the solvents (ethanol, acetone, methanol, and hexanol) employed in the extraction procedure, and also the dilution ratios [39].

### 2.3. Soxhlet Extraction (SE)

Soxhlet extraction is considered a standard technique for over a century and represents a main reference for the development of new extraction methods [40].

Several solvents or extractants (the term used to refer to the solvent used for extraction process) [40] can be used for Soxhlet extraction (SE) of phenolics, such as methanol, ethanol, acetone, diethyl ether, dichloromethane, n-hexane, and ethyl acetate [38]. The use of solvents should be limited, mainly because of their negative influence on humans’ health and environment implications. Additionally, if the extract is further utilized in the food industry, the solvent must be removed through an evaporation/concentration procedure [38]. The main shortcoming of the Soxhlet extraction method is that the extract is constantly heated at the solvent boiling point, which might have a negative effect on thermolabile compounds or, moreover, can led to the formation of artifacts, according to Seidel et al. [41]. In line with this, recently, Popovici et al. [42] showed that prolonged Soxhlet extraction (8 h at 100 °C) has diminished total phenolic content of the analyzed samples.

Even though the use of conventional SE for BSG phenolic compounds has been widely spread due to high replication and effectiveness, the main drawbacks of SE leads to sustainability and environmental problems. When compared to the LSE of BSG phenolic compounds, the SE requires less time and has a higher consistency [38].

Moreira et al. used ethanol to perform a BSG phenolics SE extraction for 4 h. The stated yield (0.0014 ± 0.0001% for ferulic acid, *w*/*w*) and the difficulties of the procedure may recommend the adoption of alternative techniques rather than SE for phenolic BSG extraction [25]. Optimizing the treatment parameters can lead to a higher yield, confirming the SE as a viable extraction method [8]. Furthermore, the use of auxiliary energies such as microwaves and ultrasound highly improved the SE method effectiveness [40].

## 3. Advanced Treatments and Extraction Techniques

### 3.1. Enzymatic-Assisted Extraction (EAE)

Enzymes used in extraction processes are derived from a variety of sources including animal organs, vegetable extracts, and fungal or bacterial samples. Cellulases, pectinases, and hemicellulases are the most common enzymes applied for the extraction of secondary metabolites. Their mechanism of action involves the hydrolysis reaction by degrading the plant material cell wall integrity, enhancing the cell wall and membranes permeability, and increasing the extraction yield of targeted bioactive compounds [18]. Most phenolic compounds, for example, are stored in an inaccessible form, frequently bounded in the cell wall of polysaccharides as cellulose, hemicellulose, and pectin (Figure 2), and connected through hydrophobic connections and hydrogen bonds [43]. Therefore, enzymatic pretreatments applied in bioprocesses aims to solubilize the cell wall of the plant producing the accelerated release of intracellular biomolecules. The enzyme classification grouped cellulase into three groups: endocellulases, exocellulases or cellobiohydrolases, and cellobiases or β-glucosidases [44]. The hydrolysis reaction begins after the enzyme complex activates on specific sections of the cellulose, culminating in the transformation of the cellulose into glucose [44].

Protein is also a key nutrient that is discarded with cereal industrial waste such as BSG. A recent study by Rommi et al. [45] presented an alkaline protease treatment protocol aiming to remove BSG protein. Protease pretreatment enhances protein extraction yield from 15% to 100%, compared to the untreated BSG sample. Because of their properties and nutritional worth, lipids are also an important compound of food composition.

Cereal by-products, such as cereal germs, can also be used to recover high-quality lipids components, as unsaturated fatty acids (43–64 g/100 g) [46]. Moreover, a study performed by Niemi Piritta et al. showed that a large amount of free fatty acids from triglycerides and phospholipids was released due to the action of lipase on BSG by-product [47]. Table 2 lists other valuable compounds that can be recovered from cereal by-products through enzyme-assisted extraction.

To apply the enzymatic pretreatments most efficiently in extraction protocols or bioprocesses, it is critical to understand the catalytic property of enzymes and their mechanism of action, the sample matrix’s cell wall structure, along with the optimal operational parameters, and the best enzymes selection for the plant material analyzed [16,18]. The enzyme complex utilized for a certain matrix must be chosen based on the specific functionality of the enzymes according to the cell wall chemical structure. Furthermore, optimal parameter conditions such as process time, temperature, pH, and enzyme concentration are some of the important criteria that must be satisfied for enzyme pretreatments aiming to maximize secondary metabolite accessibility. Each enzyme has an optimal temperature for maximum catalytic activity, but it still provides flexibility for larger temperature range, and for most of them it is in a range between 30 and 60 °C [48]. Apart from temperature, extreme pH conditions can also affect enzyme activity [48]. To conclude, we can assess that the main factors involved in the release of bioactive compounds generated through an enzymatic pretreatment must be optimized for each specific process and for each source material.

**Table 2 foods-11-02454-t002:** Summary of EAE (*Enzymatic-assisted extraction)* for the extraction of bioactive compounds from cereal by-products.

Source	Targeted Compound	Enzyme	Commercial Formulation	EAE Parameters	Results	Application	References
Rice bran	Fatty acids	Alcalase	Alcalase 2.4 L by Novozymes Bagsvaerd	Powdered rice bran was mixed with distilled water at a ratio of 1:7.5 (*w*/*v*); pH mixture: 9.0; Temperature: 57 °C; Time: 150 min; Enzyme quantity: 2 g/100 g	Higher content of unsaturated fatty acids: 76.31%; Tocopherols and tocotrienols: 1004 mg/kg; Sterols: 7749 mg/100 g; Squalene: 2962 mg/kg; Oryzanol: 2.43 g/100 g; Extracted oil has lower crystallization and melting points. The wax and phospholipid concentrations of the extracted oil were reduced.	Edible oil in food industry	[49]
Rice bran	Protein	Trypsin type I	Trypsin type I from bovine pancreas	50 mL of protein solution was hydrolyzed using a ratio 1:100 enzyme-substrate; Temperature: 37 °C pH: 8 Time: 4 h; Enzyme activity: 10,000 BAEE units/mg of protein. The enzymatic activity was inactivated by reducing the final pH to 3.	Protein concentration (g of protein/100 g of extract): Albumin: 42.40 ± 1.70 Globulin: 41.14 ± 2.72 Glutelin: 69.01 ± 1.53 Total protein soluble: 63.20 ± 1.70	Foods, cosmetics and pharmaceuticals	[50]
Sesame bran	Protein and phenolics	Alcalase	Alcalase 2.4 L by Novozymes	Enzyme concentration: 0.12–2.40 AU/100 g Ratio sesame bran and dH_2_O: 1:10 (*w*/*v*); pH: 9.8; Temperature: 45 °C; Time: 30 min; Vacuum time: 1–30 min; Vacuum pressure: 100–650 mmHg.	Combined enzymatic treatment resulted in 19.1% and 61.4% more protein yield; Increased protein yield, total phenolic content and antioxidant capacity values.	New extraction protocols including vacuum treatment.	[12]
Corn husks	Flavonoids	Cellulose	EC 3.2.1.4 by Macklin Biochemical Co., Ltd.	Extraction solvent: aqueous ethanol; Enzyme dosage: 0.3–0.5 g/100 g; Incubation time: 1.5–2.5 h Liquid-to-solid ratio: 30–40 mL g^−1^; Temperature: 40 °C; pH: 5.0; The enzyme was inactivated in boiling water for 5 min	1.3 g/100 g of total flavonoids of dry waste were recovered;	Corn industry	[51]
Brewer’s spent grain	Arabinoxylans	Xylanases; Peptidase.	EC 3.2.1.8 by AB Enzymes; Clarex by DSM Food Specialties.	2 and 5 units of xylanases; 25 µg of the peptidase; Temperature: 50 °C; Time: 15, 45, 90, 150 and 240 min.	Over 33% of Arabinoxylans was solubilized.	Food ingredient	[52]
Rye bran	Phenolic acids	Xylanase; Amylase.	Grindamyl A 1000 Depol 740 L	200 nkat/g bran xylanase; 5 nkat/g bran amylase 65% water content; Temperature: 40 °C Time: 4 h	Ferulic acid production was greatly improved by the applied bioprocess. Reduced levels of phenylacetic acids was were identified.	Food product (bread)	[53]
Brewer’s spent grain	Dietary fiber, protein, unsaturated fats, and lignans	Xylanase; Alcalase	Depol740 L; Celluclast Alcalase 2.4 L by Novozymes	First hydrolysis: pH: 5.4 Time: 5 h Temperature: 50 °C; Second hydrolysis: pH: 10 Time: 4 h; Temperature: 60 °C.	Solubility rate: 66% of BSG; Lipids content: 11% The main fatty acids identified: linoleic, palmitic, and oleic acids; The most abundant lignans: syringaresinol and secoisolariciresinol.	Food ingredient	[47]
Brewer’s spent grain	Carbohydrates	Cellulase-hemicellulase mixtures	Econase; Spezyme CP; Depol 740 and 686.	Time: 5 h; pH: 5; Temperature: 50 °C; The enzymes were dosed according to their xylanase activity.	Carbohydrates solubilization: 26–28%; Arabinoxylans solubilization: 30–34%; Due to the presence of feruloyl esterase activity in the enzyme cocktail, released ferulic acid, arabinoxylan-oligosaccharides, and their monomers were produced; The unhydrolysed fraction contains over 40% of carbohydrates.	Food and non-food application	[54]
Brewer’s spent grain	Protein and lignin	Protease	Biotouch Roc 250 L	pH: 10; Time: 5 h; Temperature: 50 °C; Enzyme inactivation was performed by boiling the tubes for 10 min.	Increased protein solubilization from 15% to almost 100%.	Valorization of BSG into multi-use food ingredients	[45]
Rice bran	Antioxidant peptides	Proteases (papain, flavourzyme, neutrase, protamex, and trypsin)	Novo Nordisk Co	Time: 3 h; Temperature: 37–55 °C (optimal for each enzyme); pH: 6.5–8.0 (optimal for each enzyme).	Highest antioxidant activity was performed by papain and flavourzyme activity;	Suitable natural antioxidants for food processing and ingredient for functional foods.	[55]

### 3.2. Ultrasound-Assisted Extraction (UAE)

Ultrasound is constituted of mechanical sound waves with extremely high frequencies that the human ear cannot detect [56]. It is considered a promising technology in the food processing industry, including the recovery and reuse of cereal waste and by-products, because it creates chemical, biochemical, and mechanical changes in liquids and gases as a result of intense cavitation and the generation of high intensity acoustic fields.

UAE improves significantly the optimal bioactive compounds recovery by requiring less time, energy, and solvents than conventional extraction methods, with the added benefit of using low temperatures and ensuring high extraction yields for temperature-sensitive compounds. It is applied in the industry because it is easy to use, efficient, and requires lower solvents than other industrial methods for extracting bioactive compounds [57]. Several mechanisms have been identified as being involved in UAE extraction such as: fragmentation, sonoporation, sono-capillary effect, erosion, destruction-detexturation of plant structures, and local shear stress. The increased extraction yields might be explained by the physical effects of ultrasound on the raw material [58].

Commonly ultrasonic systems are: ultrasound bath, ultrasound reactor with stirring, ultrasound probe and continuous sonication with ultrasound probe. Conventional or modern methods have been combined with ultrasounds in order to obtain better extraction yields: ultrasound-assisted Soxhlet extraction, ultrasound-assisted Clevenger distillation, combination of UAE and microwave-assisted extraction (MAE) by means of simultaneous irradiation, combination of instant controlled pressure drop (DIC) process and ultrasound, combination of ultrasound and supercritical fluid extraction, combination of ultrasound and extrusion extraction, ultrasound-assisted enzymatic extraction, and ultrasound-assisted emulsification microextraction [59,60].

#### 3.2.1. Parameters That Influence UAE Process

According to Rutkowska et al. [61], parameters which might affect the acoustic cavitation and extraction process are the following: ultrasound frequency, where the most commonly used ultrasonic waves are in the range of 20 to 100 kHz, ultrasound power which is directly corelated with UAE efficiency (regarding yield and composition of the extracts), ultrasonic intensity which increases the sonochemical effects and UAE efficiency, physical parameters of the solvent such as viscosity, surface tension, vapor pressure, and the solubility of the target compounds. Temperature is also an important parameter that should be optimized to obtain the highest extraction yield without degradation of the bioactive compounds. Additionally, the matrix pretreatment such milling, drying, and flaking is important and has an impact on the extraction efficiency.

#### 3.2.2. UAE Applicability in Cereal Waste Valorization

UAE is commonly used to extract proteins from a variety of agricultural sources, including soy, sorghum, and defatted rice bran [62]. Cereal solid wastes might be a low-cost resource for recovering phytochemicals, with potential use in the pharmaceutical, cosmetic, and food sectors [63]. The impact of ultrasound on innovative techniques, such as the controlled release of encapsulated peptides, have been studied. Ultrasound treatment prior to enzymatic hydrolysis, for example, has a stimulatory effect and may be effective in obtaining protein hydrolysate rich in short chain bioactive peptides. Polyphenols are another family of chemicals found in large quantities in cereal solid wastes. Polyphenols may provide cardiovascular advantages due to their antioxidant activity, as well as antibacterial, antiviral, and anti-inflammatory properties [64,65]. UAE is considered to be more effective than traditional extraction for the total polyphenols BSG recovering, according to Alonso-Riano et al. The particle size and solvent type were the most significant factors [6]. Table 3 presents some of the fractions recovered using UAE from cereal waste and by-products.

### 3.3. Microwave-Assisted Extraction (MAE)

Microwaves are electromagnetic waves that consist of a magnetic and an electronic field which oscillate perpendicular to each other at different frequencies (ranging from 300 MHz to 300 GHz) [74]. The waves act upon a material, such as cereal bran, able to absorb a part of the electromagnetic energy and transform it into caloric energy by two mechanisms: ionic conduction and dipole rotation. Ionic conduction is defined as the migration of ions and electrons under the influence of the microwave’s electric field, resulting in friction and heating. When dipolar molecules try to align with the electric field in the presence of microwaves, their oscillation produces heat [75,76]. MAE (Figure 3) is a feasible method for extracting chemicals with medium or high polarity from solid matrix such as cereal by-products (bran, germ, husk, and pericarp) since it is dependent on the dielectric constant of both the solvent and the matrix. The solvent should be polar in order to heat under the action of microwaves, generating internal heating and cell disintegration, allowing for the separation of the concerned molecules. MAE is ecologically sustainable since it uses less energy and a diverse selection of non-toxic solvents [77,78].

MAE extraction systems are classified into two types: closed and open. Extractions in a closed system are carried out in a sealed vessel using uniform microwave heating. The extraction is fast and efficient due to high working pressure and temperature, but is susceptible to losses of volatile compounds, with limited sample throughput. Open systems were designed to fix the drawbacks of closed systems and are considered more suitable for extracting volatile compounds, such as polyphenols, which are commonly found in grain waste. In an open system, only a part of the vessel is directly exposed to microwave radiation while the upper part of the vessel is connected to a reflux unit to condense the vaporized solvent [79]. MAE systems have been incorporated into other extraction systems, in an effort to improve extraction efficiency and ensure a greener implementation. Microwaves and negative pressure cavitation, ultrasonic microwave-assisted extraction, microwave-assisted subcritical and supercritical fluid extraction, microwave-assisted enzymatic extraction, microwave hydrodiffusion and gravity extraction are examples of MAE associations which led to better extraction yields, faster extraction times, minimal degradation of components and energy saving [80].

#### 3.3.1. Parameters That Influence MAE Performance

According to Angiolillo et al. [81], there are few parameters that influence the extraction process, as follows:

*Solvent*—Three characteristics should be considered: the microwave-absorbing properties (dielectric constant), the interaction with the matrix, and the analytic solubility of the chosen solvent. The amount of solvent used usually ranges between 10 and 30 mL. The organic solvents used in cereal wastes valorization are aqueous solutions composed of ethanol, methanol, or acetone [82].

*Temperature and pressure*—are important factors contributing to increased recoveries of the analyte. Their values depend on the chosen cereal waste or byproduct (either by itself or pretreated), the analyte needed to be recovered (volatile analytes such as phenolic compounds, furfural, and levulinic acid may degrade at high temperature), solvent used (if the analyte dissolute in it) and the system type (open or closed) used for MAE.

*Extraction time*—is very short compared to conventional extraction methods. Depends on the analyte needed to be extracted (some compounds from cereal waste may degrade during long extraction times such as volatile compounds), the matrix from which the analyte is extracted and the chosen temperature, with a range between 5 and 30 min.

*Matrix nature*—plant matrices have a lot of water, but cereal wastes usually need to be pretreated before MAE (milled, grinded, or frozen and mixed with water/extraction solvent).

*Other*—power and stirring are also parameters that help increase the yield of the MAE process.

#### 3.3.2. MAE Applicability in Cereal Waste Valorization

Microwave-assisted extraction is a novel technique that provides a low degradation rate of compounds [83]. In Table 4, the principal fractions recovered from cereal waste using MAE are summarized.

### 3.4. Membrane Fractionation of Different Compounds from Cereal Waste and by-Products

The recovery of different classes of nutraceuticals, including lipid based, carbohydrate based, protein based, and polyphenols can be done using membrane technology. The technology’s great selectivity allows for the expansion of a new biorefinery idea. Thus, agro-industrial waste and by-products may be used to produce natural antioxidants with nutraceutical value, as well as macromolecules such as biopolymers and biofuels (i.e., bioethanol and biogas). The product quality, environmental impact, plant compactness, and energy saving are improved due to the possibility of integrating various membrane operations in the developing of these technologies [4].

Membrane separation technologies were successfully used in the agro-industrial process, in order to obtain food, pharmaceutical and biotechnological products [93]. Different types of membranes, materials, and configurations were utilized to recover bioactive compounds from food waste, such as polysulfone, polyethersulfone, and composite fluoropolymer [94]. The process of membrane separation is described in Figure 4. The extraction is performed in a vial where the aqueous phase is separated from the organic phase through a flat membrane. According to the partition coefficient in the sample–solvent mixture, the targeted compound crosses the membrane to the acceptor phase. In order to prevent the loss of solvent through the membrane, nonpolar solvents are recommended to be used [20].

Membrane separation technology is used for recovery, fractionation, and concentration of target compounds from products, by-products, and wastes, regardless of aqueous and alcoholic processing streams [4]. Due to its low operating costs and superior product quality compared to the conventional methods such as Soxhlet extraction, this technique has earned a significant role in the separation, concentration, and purification of phenolic compounds [4]. The advantages of membrane separation technology include functionality, environmental friendliness, energy savings, and high product quality. Furthermore, the quantity of solvent used for exaction is quite low (e.g., approximately 800 µL), according to [20]. Therefore, due to its advantages, recently, pressure-driven membrane process based on ultrafiltration, microfiltration, and nanofiltration have performed increasing importance in the agro-food sector [20].

On the other side, the membrane bio-separation method involves a number of limitations, such as the high cost of the fixed membrane area, but this drawback could be neglected considering the high purity of the obtained product and the ability for recovering of a large number of by-product components [93]. The separation process performance is influenced by several factors such as the selectivity of the membrane and its flux permeability which are further corelated with operating parameters (pressure, configuration of the process, temperature, cleaning protocol or even module characteristics). It is also important to mention that membrane characteristics play a key role in the separation process, mainly its pore size, material, and structure [93].

Considering the aforementioned factors involved in the separation performance, many membranes are applied in the purification processes rather than in the extraction technology [20,93]. Due to their properties and advantages over conventional technologies, membrane technologies provides an excellent potential for resource recovery in the field of cereal processing by-products [4].

Filtration, including microfiltration and ultrafiltration was a frequently used technique aiming to separate fractions depending on their molecular size. For example, the membrane separation was used to obtain the solid fractions from corn. The high protein corn gluten meal (67% protein), high-fat corn germ, corn starch, and high fructose corn syrups were isolated. Another example is the use of ultrafiltration to recover proteins from brewers’ spent grain. A BSG extract was prepared and ultrafiltered through two categories of membranes, 5 and 30 kDa, and over 92% of the protein was retained. In the final product, the percentage of protein contents was 20.09% when 5 kDa membrane was used and 15.98% in case of 30 kDa membrane. Since there is no additional heat treatment, this method of fractionation yields to a high-quality protein [95].

The recovery of high added-value compounds from different agro-food by-products such as olive mill, artichoke, wastewaters, citrus by-products, soy processing waste stream, grape and wine by-products, vegetable aqueous extracts, and whey processing using integrated membrane process was efficiently studied on laboratory and pilot scales [4]. Developing integrated membrane procedures can lead to sustainable agricultural and food production as well as other industrial areas.

### 3.5. Pressure Based Extraction Techniques

#### 3.5.1. Supercritical Fluid Extraction Principle and Characteristics

Supercritical fluid extraction is defined as the separating process of a matrix interest compound using as extracting solvent, one supercritical fluid (Figure 5). A supercritical fluid can diffuse through solids like a gas and dissolve materials like a liquid, being a substance at a temperature and pressure above its critical point. Small changes in pressure and temperature influence the density of the supercritical fluid, contributing to the dissolving power [96,97].

It is an environmentally friendly technology used for the extraction of bioactive compounds from cereal wastes such as catechin, epicatechin, flavonoids, polyphenols, procyanidin, and tocopherols [98,99]. One of the most commonly used solvent is CO_2_ due to its cost-efficiency and capacity to be easily removed; its critical temperature of 31.1 °C and pressure of 7.3 MPa makes it ideal for processing volatile compounds and it is extensively used in food industry because of its non-corrosive, non-toxic, colorless, and odorless properties [100].

Due to the fact that the supercritical temperature is close to the ambient temperature, supercritical fluid extraction based on carbon dioxide (SFE-CO_2_) has a low oxidative and thermal impact with multiple benefits for the extracted compounds such as oils from cereal waste: high quality, unaltered original properties, and no contamination by residual liquid solvents [101]. When extracting high polar compounds, such as polyphenols from barley hulls, the CO_2_ system needs the addition of a co-solvent to increase the solubility in the supercritical solvent. Ethanol and methanol are co-solvents which were used for increasing extraction yields of catechin, epicatechin, and gallic acid, but ethanol is preferred because of its better suitability in the food industry [102].

Machado et al., 2013, describes the operational system used for the extraction with supercritical fluids. The preparative system, used in pilot and industrial scales, which is able to extract grams or kilograms of compounds consist of a pump for the solvent, a pump for the co-solvent (if necessary), an extraction cell or column equipped with independent temperature and pressure coolers and one or more separators needed to collect the extract and the depressurized solvent. During the process, the separators can be placed in series with different conditions applied in order to separate different compounds at different parameters. Various accessories can be added such as a cooling system for capturing the volatile compounds or an automatic sampler [103].

The advantages of using supercritical fluid extraction (SFE) for extracting certain compounds such as antioxidants from cereal waste are: high yields of extraction, specificity toward targeted compounds, no additional separation steps and maintaining the chemical structure and functional activity of targeted compounds. The main disadvantage is the equipment needed, which is expensive, compared to other equipment needed in other extraction methods [104,105]. Supercritical fluid extraction of interest bioactive compound from a solid raw material may involve three different stages: internal mass transfer, phase equilibrium and external mass transfer. Reboleda et al. mentioned that oil extraction yield may be highly affected by operational parameters such as solid pretreatment, extraction pressure and temperature and solvent flow rate [106].

In order to optimize the SFE, knowledge about solvent thermodynamic data (solubility and selectivity) and kinetic data (mass transfer coefficients) are needed. The extraction curve is the kinetic representation of SFE, and it is represented in a graphic of extracted mass versus extraction time and depends on the process parameters (solvent flow rate, bed particle size). The study of extraction curves and knowing the effects of the variables during the extraction allow the solvent flow rate and extractor capacity to be established [107]. The characteristics of the supercritical fluid are important: density, solvent power, viscosity, diffusivity and low value or absence of surface tension provide high solubility and selectivity, allowing rapid penetration into the sample matrix [108]. Several parameters impact the efficiency of the extraction defined in terms of recovery of the compounds of interest and extraction yield as follows:

*Temperature*: needs to be chosen at a higher value than the one corresponding to the critical point (Tc—critical temperature—characteristic for a given substance). Two effects were observed when increasing the temperature and maintaining the pressure: density effect (an increase in temperature decreases the density of the fluid and decreases solubility) and volatility effect (increasing the temperature increases the volatility of the solute, increasing solubility). Depending on the crossover pressure, the density effect or the volatility effect may have a greater impact on solubility behavior [109]; when extracting polyphenols, the temperature is critical, and thermolability of the compound of interest must be considered: at a higher temperature, extraction of thermostable compounds offers higher yields and lesser extractions time, while on thermosensitive compounds, it harms the extraction yield and stability of volatile compounds such as flavonoids;

*Pressure*: needs to be chosen at a higher value than the one corresponding to the critical point (Pc—critical pressure-characteristic for a given substance). A higher pressure applies a pressure effect upon the density of the solvent (CO_2_), improving solubility [20,110];

*Time*: in order to establish the extraction time, knowledge of the transport properties of the SCF solvent: viscosity, diffusivity and thermal conductivity, is required. In general, the time required for extraction should be as brief as possible while maintaining the qualitative composition of the extract. It decreases when pressure increases or when temperature decreases at relatively lower pressures;

*Solvent-to-feed ratio*: a high fluid to-solid ratio increases the extraction rate and reduces required extraction time. An optimum fluid-to-solid ratio is also determined by other factors, such as the nature of the plant material, sample fragmentation, or the quality of the final product [20];

*Particle size*: extraction rate and yield increase with decreasing particle size. This could be explained by the fact that larger particle dimension leads to higher extraction time, and therefore to a smaller yield. However, a smaller particle size may result in a higher drop in pressure associated with filter cake formation and consequently leads to a reduction in the mass transfer rate [111];

*Cosolvent addition*: needed when the compound of interest has high polarity. Supercritical extraction is primarily used to isolate nonpolar bioactive compounds such as carotenoids and lipids because the solvents used in this technique, including CO_2_, are non-polar. For the extraction of polar compounds, such as flavonoids, the addition of modifiers such as ethanol, methanol, or water are highly required [112].

Subcritical and supercritical extraction methods (pressurized liquid extraction-PLE, gas-expanded liquids extraction-GXL and supercritical fluid extraction- SFE) are efficient in extracting bioactive compounds from natural sources [113,114]. Bioprocessing used SFE with different applications, such as: extraction of fermentation products, bio-oil production, production of pharmaceutical ingredients from various wastes, including cereal waste, removal of biostatic agents, and organic solvents from fermentation broth, SCF disruption of yeasts, and treatment of lignocellulosic materials [115]. Table 5 presents some of the recovered fractions from various cereal waste, mentioning the extraction parameters and the applicability of the bioactive compounds extracted.

#### 3.5.2. Pressurized Liquid Extraction (PLE), Subcritical Water Extraction (SWE) and Steam Explosion (STE)

Pressurized liquid extraction (PLE), is defined as an advanced technique which combines high temperatures and pressures below their critical points, keeping the solvent in the liquid state during the entire extraction process [128]. The advantages of using PLE are increased mass transfer rate, lower viscosity and solvent surface tension and improved analyte solubility at higher temperature [129]. Furthermore, PLE represents a faster extraction process with a lower solvent consumption, with automated system which enhance the reproducibility and facilitate the laboratory work [128]. Moreover, the method allows for the use of green solvents such as ethanol, D-limonene, and ethyl lactate which are considered environmentally friendly ones [128]. Pazo-Cepeda et al. used pressurized hot water extraction and pressurized aqueous ethanol as solvents for extraction of ferulic acid (FA) and other phenolic compounds from wheat bran obtaining maximum FA extraction yield with PHW method at 160 °C and 74 min extraction time [130]. Moreover, Dunford et al. [131] showed that PLE could be successfully used for wheat germ oil extraction, a by-product of the wheat milling industry, whilst, Povilaitis et al. [132] highlighted that PLE could be used for evaluating the antioxidant potential of rye and wheat brans. For germ oil extraction, authors used a temperature ranging between 45 and 135 °C at 150 psi and showed that ethanol solvent enhanced a better extraction yield, while a pressure of 10.3 MPa, 80 °C, and methanol–water (80:20%) solvent led to a higher extraction yield of rye and wheat brans, respectively.

The shortcoming of this method could be the use of high temperatures (in the range of 100–300 °C) and high-pressure vessel for treatment [133].

Subcritical water extraction (SWE)—uses water as a solvent but maintains its liquid state at a critical temperature between 100 and 374 °C and under the critical pressure (1–22.1 MPa). Dissolution properties of water are modified, being similar to organic solvents, thus allowing the dissolution of medium or low polarity compounds [134]. Freitas et al., 2021, mentions the advantages of SWE: it is an environmentally friendly method with water used as solvent, is easy to perform, with few extraction steps, and it has low maintenance costs; however, the main drawbacks could be the degradation of thermolabile compounds and the potential of oxidizing or catalyzing the hydrolysis of some compounds at elevated temperatures and pressures [135].

Rudjito et al., 2019, extracted feruloylated arabinoxylan (AX) from wheat bran, with destarching pretreatment, obtaining maximum extraction yields in 60 min extraction at 160 °C in a pilot scale SWE, suggesting the possibility to extract feruloylated AXs in large quantities which is fundamental for industrial development [136]. Pourali et al. used subcritical water as treatment medium for hydrolysis of rice bran biomass and identified significant levels of soluble sugars in aqueous solution and eleven phenolic compounds from decomposition of rice bran: caffeic, ferulic, gallic, gentisic, p-coumaric, p-hydroxybenzoic, protocatechuic, sinapic, syringic, vanillic acids and vanillin [137]. Recently, Yilmaz-Turan et al. showed that the recovery of proteins and feruloylated arabinoxylan from wheat bran could be successfully done through the use of a cascade process, where subcritical water extraction plays a key role [138].

Steam explosion (STE)—it is a three-step process in which structural components of the sample are broken down by steam heating (thermal step), moisturizing, and decompression which leads to shearing (mechanical step) and hydrolysis of the glycosidic bonds (chemical step), all with modulations of temperature and pressure. It is used as a pre-treatment technology for extracting lignocellulosic biomass from various wastes (such as cereal straw, sugarcane bagasse, pineapple leaves) [139,140]. Liu et al. used steam-explosion-assisted extraction to release the bound phenolic acids in wheat bran; ester linkages between ferulic and arabinoxylans were broken and a large amount of soluble ferulic acid was released proving that steam explosion could be a powerful method to extract phenolic compounds from wheat bran which would bring improvements to the nutrient utilization of wheat bran as an antioxidant ingredient in food industry [141]. Furthermore, according to Kong et al., the SE method is able to increase the bioactive compounds of wheat bran such as flavonoids, soluble dietary fiber, antioxidant activity, and phenolic compounds and inhibit the rancidity of wheat bran, therefore increasing its shelf life during storage. At the same time through STE, phytic acid of wheat bran is diminished and insoluble dietary fiber is converted into soluble ones [142]. Considering that STE technology is mainly based on high-pressure saturated steam, it can be concluded that the STE process has economic and ecofriendly advantages in the food and by-products industry [142].

#### 3.5.3. Pulsed-Electric Field Extraction (PEF) and High Voltage Electrical Discharge (HVED)

Among new environmentally friendly, non-thermal technologies used for extraction, pulsed electric field (PEF) and high voltage electrical discharge (HVED) are on a rising trend. Both have low energy consumption, keep the thermolabile components in food waste intact and can increase the extract’s yield and quality, but while the PEF is more and more used for food preservation in industrial cases, HVED remains at laboratory scale even if it has higher potential.

In PEF, direct-current high-voltage pulses flow on short bursts through two electrodes, crossing the treatment chamber with the sample and generating an electric field. By adjusting the electric field’s strength (through voltage or distance between electrodes) or frequency of the pulses and treatment time, electroporation occurs, which increases cell permeability and support the extraction process (Figure 6). Due to ions polarization across the membrane, the formed pores expand further than 0.5 nm radius, generating pressure on the electrically non-conductive membrane to the point of rupture, after which the cells cannot return to the original form [143].

In the last century, applications of pulsed electric field (PEF) have been successfully used for a significant increase in the yield extraction of bioactive compounds from agro-industrial by-products, mainly as pre-treatment; for instance, polyphenols from flaxseed in batch processes [144], polysaccharides from corn silk in continuous flow [145], phytosterol, germ oil from maize germ and hull in batch chamber [146], carbohydrate, protein, starch and reducing sugar from BSG [147], total free phenolic compounds, flavan-3-ols, flavonoids, and phenolic acid derivatives from BSG [148].

In HVED technology, a high-voltage electrical discharge is applied between two electrodes placed in liquid, generating a plasma channel that causes an electrical breakdown in the sample (Figure 6), damaging the cell structure and increasing membrane permeability [149,150,151,152,153]. According to Boussetta and Vorobiev, processes such as high amplitude pressure waves of, turbulence, bubble cavitation, and reactive species occur, complementing the electrical damage and facilitating the fragmentation of cell tissues and the subsequent extraction of high-added value compounds [144]. The HVED differs from PEF in terms of the geometry and the material composition of the electrodes. Compared to other emerging technologies (PEF, UAE, MAE, etc.), researches showed that HVED required lower energy for the extraction of biomolecules, less time and solvent use and low diffusion temperature [154]. The recent laboratory advances divided the technology into three categories: batch, continuous, and circulating extraction, all having similar mechanisms.

In the batch system, the needle electrode generates a high intensity electric field, and the electrical discharge materializes when the voltage is high enough, extracting the compounds independently in a batch treatment chamber. The continuous HVED extraction system uses a high voltage stainless-steel electrode and a grounded similar counterpart to generate the extraction system in a continuous treatment chamber through a generated converged electric field (using parallel disc mesh electrodes) or an annular gap (using annular electrodes). The third circulating extraction system consists of a high voltage pulsed generator, a treatment chamber, an extracting reservoir, and a transport element which enables higher extraction output and smaller chamber sizes. The high voltage stainless steel electrode (needle) is placed in the center of the grounded electrode (ring) and can generate pulses in microseconds [105]. The latest research on HVED showed that it can break down organic compounds and inactivate bacteria and yeasts which generates potential applications for chemical removal of organic impurities present in liquids and for bioactive compounds extraction from cereal waste and by-products. Boussetta et al. [144] found that HVED significantly increased polyphenol extraction from crushed and uncrushed flaxseed cake, while [149] observed a significant increase in the protein content and polyphenols in the extracts from sesame cake obtained after applying HVED and PEF compared to control samples. In line with this, several studies emphasized the efficiency of HVED technology in extracting several nutritional valuable compounds such as carbohydrates, proteins, polyphenols, or even anthocyanins, from different by-products [155,156,157,158] showed that HVED is able to led to rapeseed straws cellulose partial degradation.

Comparing the two electro technologies, HVED stands out as being a more efficient technology than PEF in terms of output and yield, due to the total damage of the cell wall, whereas the latter proved to be a more selective technique, minimizing subsequent purification steps.

## 4. Environmental and Economic Sustainability Outlook

As detailed in the previous chapters, cereal waste and by-products have a high valorization potential for bioactive compounds through multiple promising and innovative methods. They are available in high quantities, represent a disposal cost to the producers and show great potential as low-cost biomass for a biorefinery, while having a negative impact on the environment. Biorefining aims to exploit the full value of plant material by sequentially extracting and valorizing its components [159] and generating bioenergy, biofuels, and biobased chemicals and materials.

With the increased contemporary focus on circular economy regarding the industrial-sectors economic, environmental, and social aspects, a biorefinery works as a tactical instrument for the implementation of a circular bioeconomy [160]. In this context, it is clear that, in order to act in accordance with the fundamentals of the later, the bioactive compounds should be obtained through eco-friendly, sustainable, low-cost, low-carbon-footprint technologies [77]. Additionally, the use of next-generation industrial biotechnology should be viewed as a multiproduct portfolio where all the products contribute to generating revenues and also share and optimize the costs [161].

Although recent progress has been made and most of the technologies are available outside the laboratory scale, main issues still need to be further resolved. First, the large-scale industrial extraction pilot needs to be established, which helps to reduce the spoilage and economic energy consumption during the by-products storage [162]. Then, mixed use of some emerging, ideal technologies require lab scale research in order to prove the integrated efficiency, which basically require the academia, government, and economic partners (the triple helix partnership) to collaborate in a multidisciplinary approach in order to bring together their competences and achieve enhanced economic and social development on a systemic scale [163] and develop the bio-based supply chain for the greener products of the future. Third, due to the high annual quantities of cereal waste and by-products in the world, large amounts of residue will still remain after extraction. Therefore, a sustainable closed-loop system should be considered. Lastly, innovations face legislative voids or the need for approval or standardization from the regulatory authorities. These aspects represent common technical, environmental, and economic bottlenecks needed to be dealt with before they can be turned into energetically productive, environmentally sustainable, and economically feasible technologies [164].

## 5. Conclusions

Large amounts of organic waste are generated in the agro-industrial cereal processing cycle. The management of this waste creates major costs, including both properly disposing of it and reducing the environmental impact. The circular bioeconomy initiative offers a fresh perspective on food waste. Moreover, wastes from the cereal processing industries, such as by-products, can be a significant source of bioactive compounds that can be exploited in the food additive, pharmaceutical, or cosmetics industries. The extraction of bioactive compounds from organic residues involves a series of technically well-applied strategies. Conventional extraction methods are well documented, and despite their financial, environmental, and toxicity limitations, they are widely used because they can provide significant extraction yields in a short period of time. According to the findings of this paper, the use of advanced extraction methods has achieved considerable results in recent decades. Valuable bioactive compounds derived from natural sources are absolutely critical in food, pharmaceutical, and cosmetics industries, and the appropriate techniques for extracting and isolating compounds of interest from agro-food by-products necessitate comprehensive research. Therefore, the optimal extraction technique for each class of target compounds will be selected based on the extraction requirements provided by the methodology, with a direct correlation with the economic and environmental consequences.

## Figures and Tables

**Figure 1 foods-11-02454-f001:**
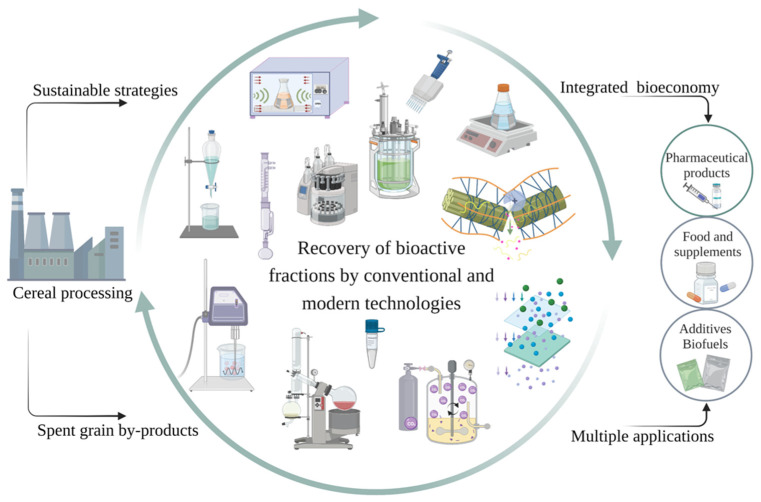
Recovery of bioactive fractions by conventional and modern technologies with different applications.

**Figure 2 foods-11-02454-f002:**
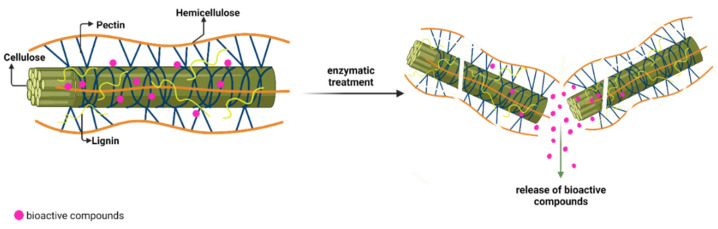
Illustration of bioactive compounds (phenolics) in the cell wall of plant structure.

**Figure 3 foods-11-02454-f003:**
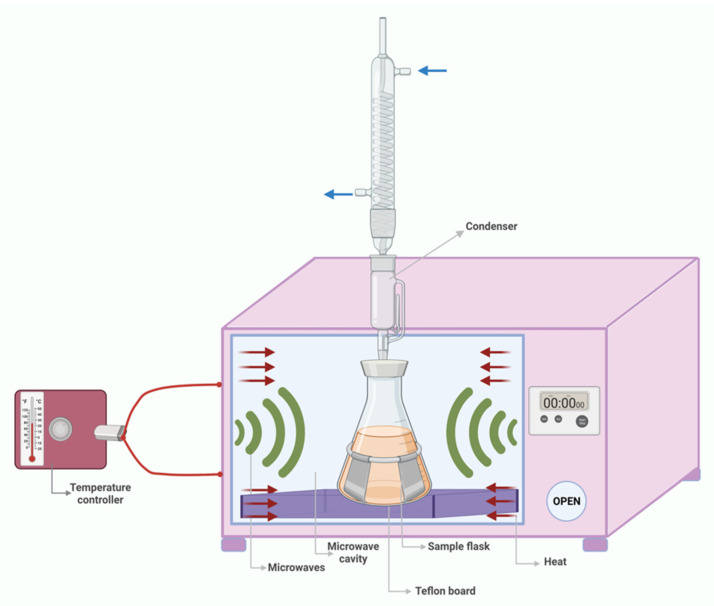
Microwave-assisted extraction.

**Figure 4 foods-11-02454-f004:**
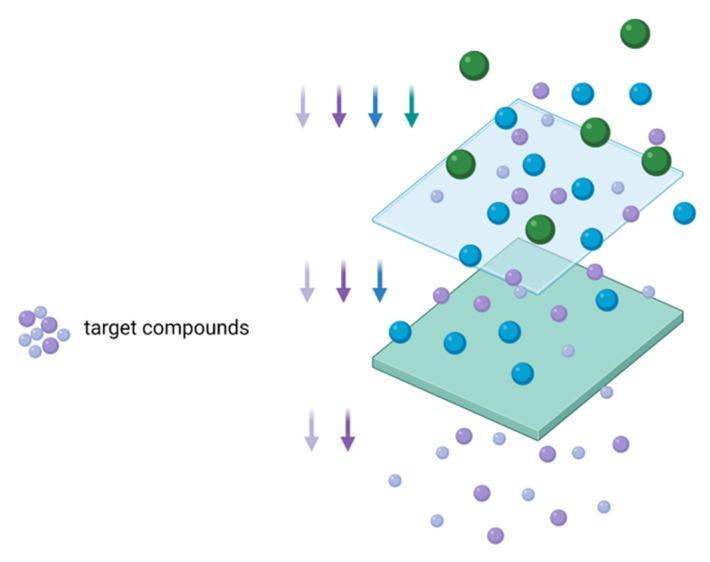
Membrane separation technology.

**Figure 5 foods-11-02454-f005:**
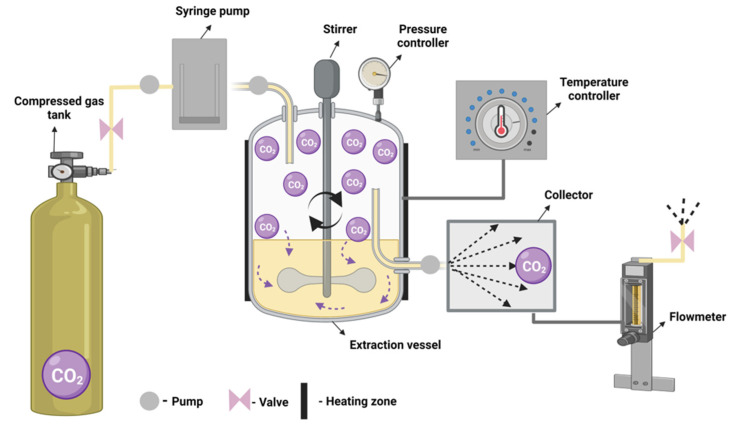
Supercritical fluid extraction.

**Figure 6 foods-11-02454-f006:**
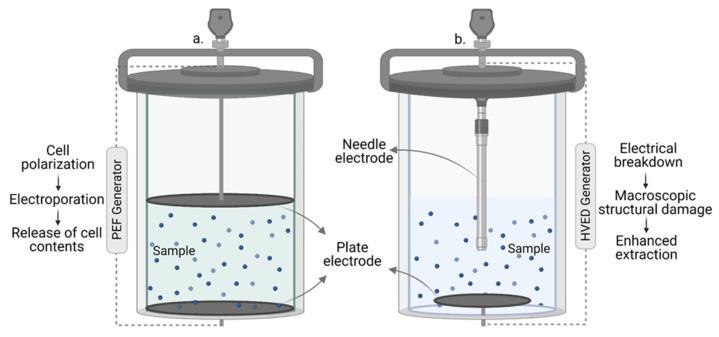
Pulsed electric field extraction (**a**) and high voltage electrical discharge (**b**).

**Table 1 foods-11-02454-t001:** Overview of phenolic compounds extracted by conventional extraction techniques.

Raw Material	Target Compounds	Extraction Method	Extraction Conditions	Yield/Released	Reference
Brewers’ spent grains	Ferulic acid	Alkaline hydrolysis	Ratio of solvent to raw material (mL g^−1^): 20 Extraction time: 90 min Temp. (°C): 110 Solvent: NaOH 2%	0.27 (% *w*/*w)*	[25]
Flax shoves	Ferulic acid *p*-coumaric acid	Alkaline hydrolysis	0.5 M NaOH 4 h at 50 °C neutralized with 6 M HCl.	25 mg/100 g 61 mg/100 g	[26]
Wheat bran				391 mg/100 g 20 mg/100 g	
Corn bran				2510 mg/100 g 350 mg/100 g	
Brewers’ spent grains	Ferulic acid *p*-coumaric acid	Alkaline hydrolysis	NaOH 2% 90 min at 120 °C	145.3 mg/L 138.8 mg/L	[27]
Yellow maize germ	Ferulic acid	Alkaline hydrolysis	2 M sodium hydroxide at room temperature for 1 h. The pH of the mixture was adjusted to 3 with 6N hydrochloric acid.	461.89 (mg FA/g dried extract)	[23]
White maize germ				522.99 (mg FA/g dried extract)	
Brewers’ spent grains	Ferulic acid	Alkaline hydrolysis	NaOH 2% (*w*/*v*) 20 mL NaOH/g 120 °C, 1.5 h	476.99 mg/100 g	[21]
Black brewers’ spent grains Pale brewers’ spent grains	Phenolic compounds	Alkaline hydrolysis	1 N NaOH for 16 h at room temperature in the dark under N_2_	5.29 (mg GAE/g dw) 3.75 (mg GAE/g dw)	[28]
Teff straw	Nanocellulose	Acid hydrolysis	Sulfuric acid concentration 44.4% *v*/*v* Time 32 min Temperature 40.5 °C	62.2% nanocellulose	[29]
Brewers’ spent grains	Hemicellulosic fraction	Acid hydrolysis	Sulfuric acid concentration 100–140 mg g^−1^ dry matter; Reaction time 17–37 min;	85.8% hydrolyzed xylan 95.7% hydrolyzed arabinan	[30]
Wheat flour and bran	Polyphenols	Acid hydrolysis	Methanol/H2SO4 90:10 (*v*/*v*); Time 20 h Temperature 85 °C	200–1600 mg/100 g polyphenols in the acidic hydrolysates	[31]
Corn fiber and wheat bran	Phenolic compounds	Acid hydrolysis	500 mL of 50 mmol trifluoroacetic acid; Time 3 h; In a boiling water bath under constant stirring.	Soluble ferulated oligosaccharides	[32]
Brewers’ spent grains	Ferulic acid	Soxhlet extraction	Ratio of solvent to raw material (mL g^−1^): 30 Extraction time: 4 h Temp. (°C): b.p. of solvent Solvent: Ethanol	0.0014 (% *w*/*w)*	[25]
Black brewers’ spent grains Pale brewers’ spent grains	Lipid content	Soxhlet extraction	70 mL analytical grade chloroform for 20 h.	9.96 (g 100/g dw) 13.51 (g 100/g dw)	[28]
Brewers’ spent grains	Phenolic compounds	Solvent extraction	Methanol Ethanol Ethanol-water 60:40 *v/v* Ethanol-water 40:60 *v*/*v* Acetone-water 60:40 *v*/*v* Acetone-water 40:60 *v*/*v*	110.58 mg GAE/100 g dw 40.97 mg GAE/100 g dw 112.04 mg GAE/100 g dw 100.38 mg GAE/100 g dw 114.23 mg GAE/100 g dw 97.38 mg GAE/100 g dw	[33]
Corn Bran	Total phenolic compounds	Solvent extraction	Water Ethanol Methanol Acetone in a water bath at 50 °C	1925 mg GAE/100 g dw 1779.5 mg GAE/100 g dw 1814 mg GAE/100 g dw 1538 mg GAE/100 g dw	[34]
Rice Bran	Total phenolic compounds	Solvent extraction	Water Ethanol Methanol Acetone in a water bath at 50 °C	1084.8 mg GAE/100 g dw 1335.9 mg GAE/100 g dw 1176.9 mg GAE/100 g dw 953.6 mg GAE/100 g dw	[34]
Brewer’s spent grains	Total phenolic compounds	Solvent extraction	Water Ethanol 80% ethanol–water 60% ethanol–water 2 min at 1900 rpm	0.51 mg GAE/100 g 0.14 mg GAE/100 g 0.56 mg GAE/100 g 0.66 mg GAE/100 g	[22]
Brewer’s spent grains	Total phenolic compounds	Solvent extraction	Water 80% Methanol 60% Ethanol 60% Acetone Hexane Ethyl Acetate 30 min at 80 °C, respectively 60 °C	3.59 (mg GAE/g BSG) 6.46 (mg GAE/g BSG) 7.13 (mg GAE/g BSG) 9.90 (mg GAE/g BSG) 4.44 (mg GAE/g BSG) 2.14 (mg GAE/g BSG)	[35]
Brewers’ spent grains light; Brewers’ spent grains dark; Brewers’ spent grains mix (light–dark, ~9:1 *w*/*w*)	Total phenolic compounds	Solvent extraction	60% Acetone 30 min at 60 ° C	2.84 (mg GAE/g BSG dw) 2.81 (mg GAE/g BSG dw) 3.85 (mg GAE/g BSG dw)	[36]

dw—dry weight.

**Table 3 foods-11-02454-t003:** The main fractions from cereal waste recovered using UAE.

Waste	Recovered Fraction	Extraction Parameters	Yield	Applicability	Reference
Wheat bran	Polyphenols	Solvent: Aqueous solution of glycerol- based eutectic mixture Temperature: 80 °C Time: 90 min Power: 140 W Frequency: 37 kHz Acoustic energy density (AED): 35 W L−1	17.78% ± 1.50	Antioxidant activity	[66]
Defatted oat bran	Phenolic compounds (TPC = total phenolic compounds)	Solvent: ethanol 80% Temperature: 70 °C Time: 25 min Power: 200–600 W Frequency: 40 kHz	TPC 184.16 mg/100 g	Antioxidant activity	[67]
Defatted oat bran	β-glucans	Solvent: ethanol 80% Temperature: 20 °C Time: 5.5 min Power: 200–600 W Frequency: 40 kHz	5.73%	Food and pharmaceutical industry, Cosmetic industry as moisturizer	[67]
Rice bran	Lactose/gluten free protein	Solvent: water (sample–water—0.5:10) Pulse on/off in the emission of power (60 s on/30 s off) Time: 10 min Temperature: room temperature	11.71%	Food formulation Protein supplementation	[68]
Wheat germ	ACE-inhibitor peptides	Solvent: water Pulsed on-time and off-time of 500 and 5 s Power: 24 W Frequency: 24 ± 2 kHz Time: 120 min	65.9%	ACE-inhibitory activity	[69]
BSG	Proteins	Solvent: NaOH 110 mM Power: 250 W Duty cycle 60% (pulsed on/off time of 3/2 s) Time: 20 min Temperature: 25 °C	86.16%	Plant-based protein source to the food industry	[70]
Red sorghum bran	Polyphenolic compounds	Solvent: ethanol 53% (52:1 mL/g of solvent to solid ratio) Time: 21 min Frequency: 25 kHz Power: 200 W	49.743 mg GAE/g dw in total polyphenols	Antioxidant activity	[71]
Durum wheat bran	Free phenolics	Solvent: ethanol 65% Time: 25 min Frequency: 48 kHz	17.29 ± 1.40	Antiradical and antimicrobial activities	[72]
Rice bran	Free phenolics	Solvent: ethanol 65% Time: 45 min Frequency: 48 kHz	19.73 ± 1.45	Antiradical and antimicrobial activities	[72]
Hull-less barley bran	β-glucans	Solvent: ethanol 80% Time: 60 min Power: 100 W Temperature: 50 °C	0.3% crude glucans	Food and pharmaceutical industry	[73]

**Table 4 foods-11-02454-t004:** The main valuable components from cereal waste recovered using MAE.

Waste	Recovered Fraction	Extraction Parameters	Yield	Applicability	Reference
Wheat bran	Phenolic compounds (467.5 μg Catechin Equivalent/g)	Solvent: methanol Temperature: 60–120 °C Time: 20 min	4.71% to 5.01%	Antioxidant capacity	[84]
Corn germ	Phenolic compounds (654 μg Catechin Equivalent/g of fresh corn germ)	Solvent: methanol Temperature: 60–120 °C Time: 20 min	2.49% to 3.51%	Antioxidant capacity	[84]
Wheat germ	Phenolic compounds (1248 μg Catechin Equivalent/g)	Solvent: methanol Temperature: 60–120 °C Time: 20 min	10.11% to 14.63%	Antioxidant capacity	[84]
BSG	Ferulic acid	Solvent: NaOH 0.75% Temperature:100 °C Time: 15 min	1.3%	Antioxidant Antimicrobial agent Anti-inflammatory agent	[25]
Wheat straw	Lignin	Solvent: H_2_SO_4_ 0.46 M Power: 602 W Stirring: yes Time: 39 min	3.4 to 11.8%	Natural binder	[85,86]
BSG	Arabino-xylans and arabinoxylo-oligosaccharides	3 sequence extraction Temperature: 180 °C Time: 2 min	62%	Prebiotic effects Antioxidant activity	[87]
BSG	Arabino-xylans	Solvent: water Temperature: 210 °C Time: 2 min	43%	Prebiotic effects	[88]
BSG	Hemicellulosic sugar	Solvent: water (5 g BSG, 50 mL water) Temperature: 192.7 °C Time: 5.4 min	82%	Butanol production	[40]
BSG	Carboxymethylcellulose (CMC)	Etherification reaction of BSG to obtain CMC Solvent: 5 mL monochloroacetic acid and isopropanol Power: 200 W Temperature: 70 °C Time: 7.5 min	1.46%	Cellulose isolation	[89]
Sorghum leaves	Reducing sugar acetic acid furfural 5-hydroxymethylfurfural (HMF) phenol	Pretreatment with NaOH and HCl solutions Power: 200–800 W Time: 2–10 min	Depends on the interested compound	Biofuels Biomaterials	[90]
Corn pericarp	Xylo-oligosaccharides	Solvent: water Temperature: 175 °C Stirring: yes Time: 18 min	70.8%	Functional food Source of bioethanol production	[91]
Black rice husk	Phenolic compounds: flavonoid, anthocyanins	Solvent: ethanol Temperature: 175 °C Stirring: yes Time: 31.11 sec	Flavonoids (3.04 mg/100 g), anthocyanin (3.39 mg/100 g)	Functional food Antioxidant action	[92]

**Table 5 foods-11-02454-t005:** The main bioactive compounds from cereal waste recovered using SFE.

Waste	Recovered Fraction	Extraction Parameters	Yield	Applicability	Reference
Rye bran	Phenolic compounds (14.62 mg GAE/g)	Solvent: CO_2_ Temperature: 70 °C Pressure: 55 MPa Time: 120 min	2.5%	Antioxidant capacity	[116]
Roasted wheat germ	Phenolic compounds (6 mg GAE phenolics/g)	Solvent: CO_2_ Temperature: 58 °C Pressure: 336 bar Time: 10 min	5.3%	Antioxidant capacity	[117]
Roasted wheat germ	Tocopherol (6.7 mg/g)	Solvent: CO_2_ Temperature: 58 °C Pressure: 336 bar Time: 10 min	100%	Antioxidant capacity, cosmetic and food industry	[117]
Purple corn cob	Phenolic compounds (290 mg EC/g) and especially anthocyanins 67 mg C3G/g)	Solvent: CO_2_ + EtOH (70%) Temperature: 50 °C Pressure: 400 bar Time: 10 min	24.4%	Antioxidant capacity	[118]
Corn germ	Oil (tocopherols)	Solvent: CO_2_ Temperature: 35–86 °C Pressure: 20–53 MPa Solvent flow rate: 4–9 kg CO_2_/h	ND	Antioxidant capacity	[106]
Wheat bran	Oil	Solvent: CO_2_ Temperature: 313.15–333.15 K Pressure: 10–30 MPa	ND	Antioxidant capacity and radical scavenging activity	[119]
BSG	BSG lipophilic fractions	Solvent: CO_2_ Temperature: 40 °C, Pressure: 40 MPa Solvent flow rate: 2–3 Standard Liters/min Time: 70 min (including 10 min of static extraction time).	5.49 ± 0.07 g/100 g	Antioxidant capacity	[120]
Rice bran	Oil (total phenolics content 3.42 mg GAE/g of oil and tocopherol 5.47 mg/g oil)	Solvent: CO_2_ + EtOH (5–10%) Temperature: 40 °C, Pressure: 40 MPa Time: 120 min	14.4 g oil/100 g	Antioxidant capacity	[121]
BSG	Phenolic compounds (0.35 ± 0.01 mg/g BSG) Flavonoids (0.22 ± 0.01 mg/g BSG)	Solvent: CO_2_ + EtOH (60%) Temperature: 40 °C Pressure: 35 MPa Time: 240 min	ND	Antioxidant capacity	[122]
BSG	Tocopherols	Solvent: CO_2_ Temperature: 313 K Pressure: 35 MPa Pretreatment: milling (particle size 0.85 mm)	ND	Antioxidant capacity	[123]
Wheat germ	Tocopherols	Solvent: CO_2_ Temperature: 40 °C Pressure: 300 bar Time: 480 min	9%	Antioxidant capacity	[124]
Wheat bran	Alkylresorcinols (1119 ± 42 lg AR/g dry bran)	Solvent: CO_2_ Temperature: 80 °C Pressure: 40 MPa Time: 120 min	34.7 mg extract/g dry bran	Antioxidant capacity stimulant/inhibitory effects on some metabolic enzymes	[125]
Oat bran	Polyphenols: avenanthramides and phenolic acids	Solvent: CO2 Temperature: 50 °C Pressure: 350 bar Time: 300 min Solvent flow rate: 15 g/min	4.6–5.3% oil from oat bran sample	Antioxidant capacity	[126]
Corn gluten meal (CGM)	Lutein	Solvent: CO_2_ + EtOH (15%) Temperature: 40 °C Pressure: 6820 psi	84.7μg lutein/g CGM	Food and pharmaceutical industries	[127]

ND = not determined; GAE = gallic acid equivalents; C3G = cyanidin-3-glucoside; EC = catechin equivalent.

## Data Availability

Data is contained within the article.

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
