# Peer review of "Cereal Waste Valorization through Conventional and Current Extraction Techniques—An Up-to-Date Overview"

_foods, 2022, doi:10.3390/foods11162454_

Round 1

Reviewer 1 Report

The manuscript is well structured and complete; however, it would be relevant if they made a plus, something extra to that of other reviews that are already available on the subject and on the techniques.

Describe the advantages and disadvantages of using those techniques on byproducts

Give examples in the waste in which the techniques of PLE, SWE, STE, PEF, HVED have already been used, because they only talk about them but not about their application to the byproducts, that would be the relevant thing. 

Author Response

Response to Reviewer 1 Comments

                                                                 Cluj-Napoca, August 10, 2021

 Manuscript ID: foods-1837869

Title: Cereal waste valorization through conventional and current extraction techniques – an up-to-date overview

Authors: Anca Corina FărcaÈ™ 1*, †, Sonia AncuÈ›a Socaci1 , Silvia Amalia NemeÈ™ 2, Liana Claudia Salanță1, , Maria Simona ChiÈ™3*, Carmen Rodica Pop1, Andrei BorÈ™a3, ZoriÈ›a Diaconeasa1, and Dan Cristian Vodnar2

The manuscript is well structured and complete; however, it would be relevant if they made a plus, something extra to that of other reviews that are already available on the subject and on the techniques. 

Response: Thank you very much for your valuable remarks and suggestions. We considered them highly appropriate, and we included all of them in the present version of the manuscript. We express our appreciation of your time and effort for the preparation of comments and very valuable suggestions. Thank you very much.

With respect to the idea to ,,made a plus’’, according to a large body of literature, this is the last study that aims to analyse comprehensively both conventional extraction techniques such as acid and alkaline hydrolysis, solvent extraction, Soxhlet extraction, along with the advanced novel treatments and extraction techniques (enzyme-assisted extraction, ultrasound-assisted extraction, microwave-assisted extraction, membrane technology, subcritical and supercritical extraction, pressurized liquid extraction, steam explosion, pulsed electric field, and high voltage electrical discharge).

For instance, (Fritsch et al. 2017)  discussed about the cellulose extraction from wheat bran and oat husk and emphasized the extraction of biomolecules from bran by using thermal treatments such as steam explosion, ultrasonication, classical chemical extractions such as acid, alkaline or hydrogen peroxide or even enzymatic digestion but did not discussed about membrane technology, subcritical and supercritical extraction, pressurized liquid extraction, pulsed electric field and high voltage electrical discharge. Moreover, recently, (Plazzotta and Manzocco 2019) in the chapter entitled Food waste valorisation discussed about cereal waste and their valorization strategies but did not mentioned about the used methods for extraction of the bioactive compounds.

Furthermore, in a recent review, (Galanakis 2022) revised the sustainable practices and applications to valorise valuable components recovered from cereal processing by-products and mentioned their health properties. Moreover, different approaches and technologies for the recovery of valuable compounds from different cereal processing by-products were treated, mentioning some recovery methods such as polysulfone membrane, combined treatment with proteases and sequential fractionation with one microfiltration, alkaline treatment and isoelectric precipitation, alkaline extraction coupled with microwave-assisted extraction. Only some emerging technology such as the use of ultrasounds, microwaves and enzymatic treatment were only mentioned, but not discussed.

Therefore, to conclude, we can assess that the present review is an up to date of the recently studies regarding the valorisation of cereal waste through modern and conventional methods, with a big importance in the field.

Point 1: Describe the advantages and disadvantages of using those techniques on byproducts

Response 1: Thank you for the comment. In the introduction (lines 92-99) we already mentioned the disadvantages of the conventional methods, compared with the modern ones. Moreover, lines 123-127 described the cons of using alkaline hydrolysis method, meantime, lines 151-156 described the advantages of using solid-liquid extraction. Furthermore, Soxhlet extraction disadvantages are presented in lines 191-197, whilst lines 275-279 described the advantages of using ultrasound assisted extraction. Lines 343-351 defined microwave- assisted extraction with closed and opened systems highlighting their shortcomings and advantages; lines 419-430 described the advantages of membrane separation technology compared with conventional methods such as Soxhlet extraction. Lines 469-479, 499-507 described the advantages of using supercritical fluid extraction for different bioactive compounds exploited from cereal wastes, lines 595-597 described the subcritical water extraction, whilst lines 625-627 emphasized the steam explosion technology. PEF, UAE, MAE methods were compared with high voltage electrical discharge in lines 659-661.

Considering your suggestion, we decided to add one more disadvantage of Soxhlet extraction, lines 195-200, as follows:

The main shortcoming of the Soxhlet extraction method is that the extract is constantly heated at the solvent boiling point, that might have a negative effect on thermolabile compounds or, moreover can led to the formation of artifacts, according to (Seidel, 2012). In line with this, recently, Popovici et al., (Popovici et al. 2022) showed that prolonged Soxhlet extraction (8 h at 100°C) have diminished total phenolic content of the analyzed samples.

Therefore, we can conclude that advantages and disadvantages of using conventional and modern extraction methods were highlighted in the present manuscript.

Thank you for understanding!

Point 2: Give examples in the waste in which the techniques of PLE, SWE, STE, PEF, HVED have already been used, because they only talk about them but not about their application to the byproducts, that would be the relevant thing. 

Response 2: Thank you very much or the observation. According to your suggestion we decided to insert the next paragraphs:

  • For PLE technique, line 588. Even if there was an example of phenolic acid wheat bran extraction by using PLE technique, we decided to add the next paragraph:

Moreover, Dunford et al., (Dunford and Zhang 2003) showed that PLE could be successfully used for wheat germ oil extraction, a by-product of the wheat milling industry, whilst, Povilaitis et al., (Povilaitis et al. 2015) highlighted that PLE could be used for evaluating the antioxidant potential of  rye and  wheat brans. For germ oil extraction authors used a temperature ranging between 45-135°C at 150 psi and showed that ethanol solvent enhanced a better extraction yield, meantime, a pressure of 10.3 MPa, 80 °C and methanol:water (80:20%) solvent led to a highest extraction yield of rye and wheat brans, respectively.

  • For SWE extraction

Thank you very much for the observation, but we already gave some examples of using SWE in by-products in lines 599-608.

  • For STE extraction

Lines 613-625 were mainly used for describe the application of STE extraction in extracting lignocellulosic biomass from various wastes (like cereal straw, sugarcane bagasse, pineapple leaves), to release the bound phenolic acids in wheat bran; to extract phenolic compounds from wheat bran which would bring improvements to the nutrient utilization of wheat bran as an antioxidant ingredient in food industry; to higher increase the bioactive compounds of wheat bran such as flavonoids, soluble dietary fiber, antioxidant activity, phenolic compounds and inhibit the rancidity of wheat bran, therefore increasing its shelf life during storage.

Thank you for asking!

  • For PEF extraction

Regarding the applicability of PEF in by-products, we already mentioned in lines 647-651, as follows:

For instance, worth polyphenols from flaxseed in batch processes [141], polysaccharides from corn silk in continuous flow [142], phytosterol, germ oil from maize germ and hull in batch chamber [143], carbohydrate, protein, starch and reducing sugar from BSG [144], total free phenolic compounds, flavan-3-ols, flavonoids, phenolic acid derivatives from BSG [145].

Thank you!

  • For HVED

Even if we mentioned about the HVED application in lines 677-680, we decided to add one more sentence, as follows:

In line with this, several studies emphasized the efficiency of HVED technology in extracting several nutritional valuable compounds such as carbohydrates, proteins, polyphenols or even anthocyanins, from different by-products (Barba, Boussetta, and Vorobiev 2015), (Xi, He, and Yan 2017), (Rocha et al. 2018). Moreover,  Brahim et al., (Brahim et al. 2017) showed that HVED is able to led to rapeseed straws cellulose partial degradation.

Thank you!

Reviewer 2 Report

The manuscript entitled "Cereal waste valorization through conventional and current extraction techniques – an up-to-date overview" is well written, but it requires some revision. Authors have claimed to isolate bioactive compounds from cereal waste, but it is not certain that only bioactive compounds will be extracted. It is appropriate to use the term “isolation of secondary metabolites” instead of bioactive compounds.

Comments:

·        Since the authors have mentioned various extraction methods for isolating metabolites, you are suggested to include schematic diagrams or graphical images of those methods for better visibility to readers. For instance, how solid-liquid extraction is used? Please include graphical representations in every extraction method discussed in the manuscript.

·        Line 42: authors have mentioned that “The use of ethanol resulted in higher TPC and antioxidant activity than methanol, water, or acetone, according to [36]”. But it is not always true, even if you have cited.

·        Please be noted that Soxhlet Extraction is typically not useful for thermolabile metabolites. Discuss it in the manuscript.

·        I have noticed several cases of unnecessary use of citations; thus, authors should reduce excessive use of citations in the manuscript.

·        I have found some grammatical issues in several places, please correct them and try to lower the similarity index from 25% to further down. 

Author Response

Response to Reviewer 2 Comments

                                                                                      Cluj-Napoca, August 10, 2021

Manuscript ID: foods-1837869

Title: Cereal waste valorization through conventional and current extraction techniques – an up-to-date overview

Authors: Anca Corina FărcaÈ™ 1*, †, Sonia AncuÈ›a Socaci1, Silvia Amalia NemeÈ™ 2, Liana Claudia Salanță1, , Maria Simona ChiÈ™3*, Carmen Rodica Pop1, Andrei BorÈ™a3, ZoriÈ›a Diaconeasa1, and Dan Cristian Vodnar2

Point 1: The manuscript entitled "Cereal waste valorization through conventional and current extraction techniques – an up-to-date overview" is well written, but it requires some revision. Authors have claimed to isolate bioactive compounds from cereal waste, but it is not certain that only bioactive compounds will be extracted. It is appropriate to use the term “isolation of secondary metabolites” instead of bioactive compounds.

Response 1: Thank you very much for your valuable remarks and suggestions. We express our appreciation for your time and effort for the preparation of comments and very valuable suggestions.

With respect to your suggestion, we made a deep study regarding the definition of secondary metabolites and primary ones, as follows:

According to (Yeshi et al. 2022) the main plant secondary metabolites are terpenoids, followed by alkaloids and phenolic compounds, defined as small molecules with diverse chemical structures and biological activities. On the other hand, carbohydrates, lipids, peptides, proteins are primary metabolites, defined as main drivers of essential life functions.

Bioactive compounds are defined such as “natural or synthetic compounds with the capacity to interact with one or more components in the living tissues and exerting a wide range of effects’’ and include polyphenols, tannins, flavonoids, flavanols, vitamins (A and E), essential minerals, fatty acids, volatiles, anthocyanins, pigments, peptides, proteins, according to (Vilas-Boas, Pintado, and Oliveira 2021).

In the present manuscript, we discussed about secondary but also about main metabolites, therefore, we decided to use a general term of bioactive compounds.

Thank you for understanding!

Comments:

Comment 1:  Since the authors have mentioned various extraction methods for isolating metabolites, you are suggested to include schematic diagrams or graphical images of those methods for better visibility to readers. For instance, how solid-liquid extraction is used? Please include graphical representations in every extraction method discussed in the manuscript.

Response 1: Thank you very much for the comment. Authors have already realized 5 representative figures of the main used extraction methods. For instance, Fig 1 illustrates the recovery of bioactive fractions by conventional and modern technologies with different applications, Fig 2. represent the phenolics compounds release after enzymatic treatment, Fig 3. illustrates microwave-assisted extraction, meantime, Fig, 4 and 5 are based on membrane separation technology and supercritical fluid extraction, respectively.

Considering the reviewer suggestion, we decided to insert one more Figure, entitled Figure 6. Pulsed electric field extraction (a) and high voltage electrical discharge (b).

Thank you!

Figure 6. Pulsed electric field extraction (a) and high voltage electrical discharge (b)

Comment 2: Line 42: authors have mentioned that “The use of ethanol resulted in higher TPC and antioxidant activity than methanol, water, or acetone, according to [36]”. But it is not always true, even if you have cited.

Response 2: Thank you very much for the comment. This is only a conclusion of the cited reference. We just give an example trying to show that the type of solvent plays a main role in some bioactive compound’s extraction. Previously, in another study we mentioned that for BSG (brewer spent grain) sample, the solvent acetone:water mixture at 60% (v/v) has the highest yield of phenolics (9.90 ± 0.41 mg gallic acid equivalents)/g dw). Therefore, we did not generalize saying that ethanol is the best solvent, it was just a conclusion of Smuda et al., 2018. Moreover, also the matrix used for extraction could be a main factor in choosing the right solvent for the extraction process.

Thank you for the comment!

Comment 3: Please be noted that Soxhlet Extraction is typically not useful for thermolabile metabolites. Discuss it in the manuscript.

Response 3: Thank you for the comment. According to your suggestion, we decided to insert in the manuscript, the next sentence:

The main shortcoming of the Soxhlet extraction method is that the extract is constantly heated at the solvent boiling point, that might have a negative effect on thermolabile compounds or, moreover can led to the formation of artifacts, according to (Seidel, 2012). In line with this, recently, Popovici et al., (Popovici et al. 2022) showed that prolonged Soxhlet extraction (8 h at 100°C) have diminished total phenolic content of the analyzed samples.

Thank you!

Comment 4: I have noticed several cases of unnecessary use of citations; thus, authors should reduce excessive use of citations in the manuscript.

Response 4: Thank you very much for the comment. Considering that this manuscript is a review and not an article with our scientific results, we consider highly important to cite all the articles that have as the main subject the cereal waste valorization by using conventional and current extraction technique.

Thank you for understanding!

Comment 5:  I have found some grammatical issues in several places, please correct them and try to lower the similarity index from 25% to further down.

Response 5: Thank you very much for the comment.

We have corrected the grammatical issues, as follows:

Line 28: we replaced pressure based with pressure-based

Table 1, line 135: we replaced hydrolysed with hydrolyzed

Table 2, line 265: minÈ™ was replaced with min; specialities was replaced with specialties; solubilisation was replaced with solubilization; flavorzyme was replaced with flavourzyme

Line 287: clevenger was replaced with Clevenger

Line 307: valorification was replaced with valorization

Lines 370, 387: valorification was replaced with valorization

Lines 573, 576, 693: pressurised was replaced with pressurized

Line 643: none was replace with non

We also modified all the references according to the new added ones.

Thank you!
